# Prediction on X-ray output of free electron laser based on artificial neural networks

Kenan Li [1] ✉, Guanqun Zhou[1], Yanwei Liu[1], Juhao Wu [1], Ming-fu Lin [1], Xinxin Cheng [1], Alberto A. Lutman [1], Matthew Seaberg [1], Howard Smith[1], Pranav A. Kakhandiki [1,2] & Anne Sakdinawat [1] ✉

Knowledge of x-ray free electron lasers' (XFELs) pulse characteristics delivered to a sample is crucial for ensuring high-quality x-rays for scientific experiments. XFELs' self-amplified spontaneous emission process causes spatial and spectral variations in x-ray pulses entering a sample, which leads to measurement uncertainties for experiments relying on multiple XFEL pulses. Accurate in-situ measurements of x-ray wavefront and energy spectrum incident upon a sample poses challenges. Here we address this by developing a virtual diagnostics framework using an artificial neural network (ANN) to predict x-ray photon beam properties from electron beam properties. We recorded XFEL electron parameters while adjusting the accelerator's configurations and measured the resulting x-ray wavefront and energy spectrum shot-to-shot. Training the ANN with this data enables effective prediction of single-shot or average x-ray beam output based on XFEL undulator and electron parameters. This demonstrates the potential of utilizing ANNs for virtual diagnostics linking XFEL electron and photon beam properties.

Recent advances in X-ray free-electron lasers (XFELs)[1–6] at world-wide facilities such as SLAC[7], SACLA[8], PAL-XFEL[9], SwissFEL[10], and the European XFEL[11] have demonstrated innovative capabilities and operational configurations that are expected to greatly impact a wide range of proposed science experiments[12]. Tunable devices such as variable gap undulators and phase shifters have been integrated into the XFEL to tailor and control the electron beam[13], opening up fresh opportunities for science experiments. However, as the number of electron beam control parameters increases, so does the complexity of accelerator optimization and tuning. This, along with the shot-to-shot variations from the self-amplified spontaneous emission (SASE) process of XFELs, make it essential to understand the relationship between the electron beam parameters and the actual X-ray beam properties delivered to a sample.

To understand this relationship, several options are possible. First, the wavefront and spectrum of the XFEL pulse can be determined computationally, though this is a challenging task due to the complexity of the underlying physics, discrepancies between real-world and computational models, and the multitude of variables and parameters involved especially with the more recent generation XFELs. Second, real-time nondestructive measurements of the energy spectral and spatial wavefront properties of the XFEL pulse delivered to a sample could also be implemented. One method to do this involves splitting the X-ray pulse into reference and experimental beams using a beam splitter and taking measurements on both beams from shot to shot. This, however, can increase experimental complexity, require additional instrumentation, which may not be feasible depending on the physical constraints of the experimental setups, and reduces photon flux. In addition, accuracy would be highly determined by the quality and performance of the X-ray beam splitter optic.

To overcome these challenges, we develop a virtual diagnostics model based on artificial neural networks (ANNs) and shot-to-shot measurement data of both electron and X-ray beam parameters. ANNs are powerful tools for modeling complex nonlinear relationships, and exploration of their utility to overcome the limitations of conventional methods for accelerator optimization, tuning, and modeling is

[1]SLAC National Accelerator Lab, 2575 Sand Hill Road, Menlo Park, CA 94025, USA. [2]School of Applied and Engineering Physics, Cornell University, 142 Sciences Dr, Ithaca, NY 14853, USA. ✉e-mail: kenan@stanford.edu; annesak@stanford.edu

underway[14–17]. The majority of machine learning models for XFELs have primarily focused only on the electron beam for tasks such as accelerator and undulator tuning and optimization[18,19], with one study incorporating X-ray spectrometer data[20]. These studies were made possible due to the single-shot diagnostics of the electron beam implemented in the XFEL. With the recent development of high-accuracy single-shot X-ray wavefront sensors for both soft and hard X-rays at XFELs[21–23] and the development of single-shot soft X-ray spectrometers based on off-axis zone plates for spectral measurements[24,25], X-ray properties can now be characterized routinely. These diagnostic tools enable us to measure the spatial amplitude and phase, as well as the spectral qualities of the X-ray beam and allow us to further combine the X-ray diagnostics data with that of the electron beam diagnostics data into a model based on ANNs.

In the following set of experiments, we modulate the electron beam parameters via different accelerator operational configurations in the XFEL, including both that of routine operations with full normal electron beams and exploration of the effect of detuning, tapering, and kicking of slotted electron beams, record the electron and X-ray beam parameters on a single-shot basis, and then train an ANN-based model using the data. Detuning plays a critical role in determining XFEL modes through the dispersion relation equation[2,3,26–29] and can excite high-order modes[30]. In conjunction with tapering of the undulators, amplification of these high-order modes are expected. Both the routine case and the specialized cases of detuning, tapering, and kicking were chosen to demonstrate and understand the utility and limitations of the ANN-based virtual diagnostics model.

## Results

These experiments were conducted at the Time-resolved atomic, Molecular and Optical Science (TMO) instrument[31] at LCLS as illustrated in Fig. 1a. LCLS was operated in self-amplified spontaneous emission (SASE) mode, producing ~530 eV X-rays at a repetition rate of 120 Hz. Data from a total of 13 XFEL configurations were recorded, 12 different configurations using the slotted electron beam, and 1 configuration representing routine operations using the normal full SASE beam. In the 12 different configurations of the slotted electron beam, an energy chirp along the electron bunch was introduced for detuning and the taper and kicking parameters were varied. A slotted foil was used to create a short, coherent spike in the electron bunch by spoiling the majority of it when incident upon the foil, leaving an ultrashort unspoiled portion through the slot in the foil[32], as shown in Fig. 1a. The unspoiled portion then produces an ultrashort XFEL pulse through lasing. The undulator sections were set to two different states: no taper and optimal taper[33], and for each of these states, the electron bunch was kicked at various locations in the undulator, $n$ sections before the final section, with $n = 0, 1, 3, 5, 7$, and 9 where 0 indicates no kicking, as illustrated in Fig. 1b. This resulted in a total of 12 different configurations. For each configuration, we recorded the single-shot wavefront intensity, phase, and spectrum, as well as electron parameters from the undulators (spectrum and wavefront were measured separately for the same 12 configurations). In addition to the slotted electron beam, the full SASE beam in routine operation was used to study shot-to-shot wavefront phase variations, with similar recordings of wavefront phase and electron bunch parameters. The X-ray wavefront was measured using a Talbot wavefront sensor, and the spectrum was recorded using an off-axis zone plate on a yttrium aluminum garnet (YAG) screen, shown in Fig. 1c. See the "Methods" section for further details on the XFEL configurations and data acquisition.

In Fig. 2, we present the average spectra, wavefront intensity, and phase for each configuration, including variations with and without taper and kicking at different points along the undulator. The results show that different configurations result in distinct spectra and wavefronts. For instance, the spectra from taper configurations exhibit a higher energy tail and reduced low energy components compared to

that of the no taper cases. The intensity also increases as the electrons are kicked further downstream. The differences among the twelve phase maps indicate the wavefront's evolution with different kick locations and taper settings.

Indeed, the experiments revealed interesting XFEL physics when certain parameters of the electron bunch and the undulators are varied. In the no-taper case, due to the fact that the electrons are continuously losing energy, the radiation spectrum is skewed toward the red-shift side. For the taper case, the taper was over-tapered to introduce a detuning to set the resonant frequency in the blue-shift side compared to the radiation frequency in the exponential growth region, i.e., before the tapered region. Thus, the microbunching will now support high-order modes according to the dispersion relation discussed below in the "Methods" Section. In our case, the donut mode is excited, as shown in the intensity plot in Fig. 2, while the spectrum shows spectral tails at high energy, as seen in the normalized spectrum plot in Fig. 2.

We conducted an investigation into the correlations between X-ray properties and electron parameters by computing Pearson correlation coefficients between recorded electron beam parameters and our X-ray measurements (e.g., Zernike coefficients for wavefront phase). As shown in Fig. 1d, we created a correlation matrix to demonstrate the relationship between electron beam parameters and Zernike coefficients. The correlation matrix highlights that electron parameters exhibit intricate correlations with the resulting X-ray wavefront. These relationships are often implicit yet complex, involving a multitude of parameters that become challenging to depict and solve through conventional methods.

ANNs can solve real-world problems, such as regression or classification, by receiving inputs, performing complex calculations, and providing outputs. To map both X-ray and electron properties, we employed a conventional multilayer perceptron (MLP) model to predict X-ray outputs based on electron parameter readings. The MLP we used in this paper is depicted in Fig. 1e and is comprised of an input layer, multiple hidden layers, and an output layer. The inputs are electron parameters and the outputs are X-ray properties like wavefront or spectrum. Electron parameters consist of readings from bunch length monitors, beam position monitors at various sections, and electron attributes such as position, peak current, bunch charge, coordinates, pulse energy, etc. The X-ray wavefront phase is represented as Zernike coefficients obtained by decomposing the phase into Zernike polynomials. The X-ray beam spectrum is represented as 50 numbers obtained through binning. See the "Methods" section for further details on model training.

We demonstrate the effectiveness of our trained models by presenting predictions for (1) different configurations from different runs with slotted electron beam varying kicking locations and taper states, and (2) shot-to-shot variation within a single run with a full electron beam. Predictions are all single shots, and the averages are calculated based on the predicted single shots. These predictions are discussed in the following subsections.

### Analysis of predictions from the slotted electron beam configurations

In Fig. 3, we present a comparison between the measured and predicted average wavefront phase in Zernike coefficients for various configurations. The measurements and predictions are nearly identical, with only minor phase differences observed. The root-mean-square (RMS) prediction error for the average wavefront phases was determined to be 0.0169 rad. Furthermore, the standard deviation of wavefront phase from case to case was found to be 0.236 rad. Based on these values, the estimated relative error for predicting average case-to-case fluctuations is ~7%. Refer to the "Methods" section for further information on the prediction error and accuracy evaluation. The model accurately captured the differences and changes in wavefront

phase caused by varying electron parameters and accurately predicted the resulting X-ray wavefront phase. With the single-shot measurements of a comprehensive collection of electron parameters, we can determine the X-ray beam wavefront phase delivered to the end station.

Similarly, in Fig. 4, we compare the measured and predicted average spectra for various configurations. There is very little difference between the two. The good agreement observed in the figures is due to the fact that they represent comparisons of the averages. The model effectively captured the differences and changes in the X-ray spectrum caused by varying electron parameters and accurately predicted the resulting X-ray spectra. For instance, kicking at a more upstream location results in more symmetrical spectrum curves, and taper leads to spectral tails at high energy, while no taper

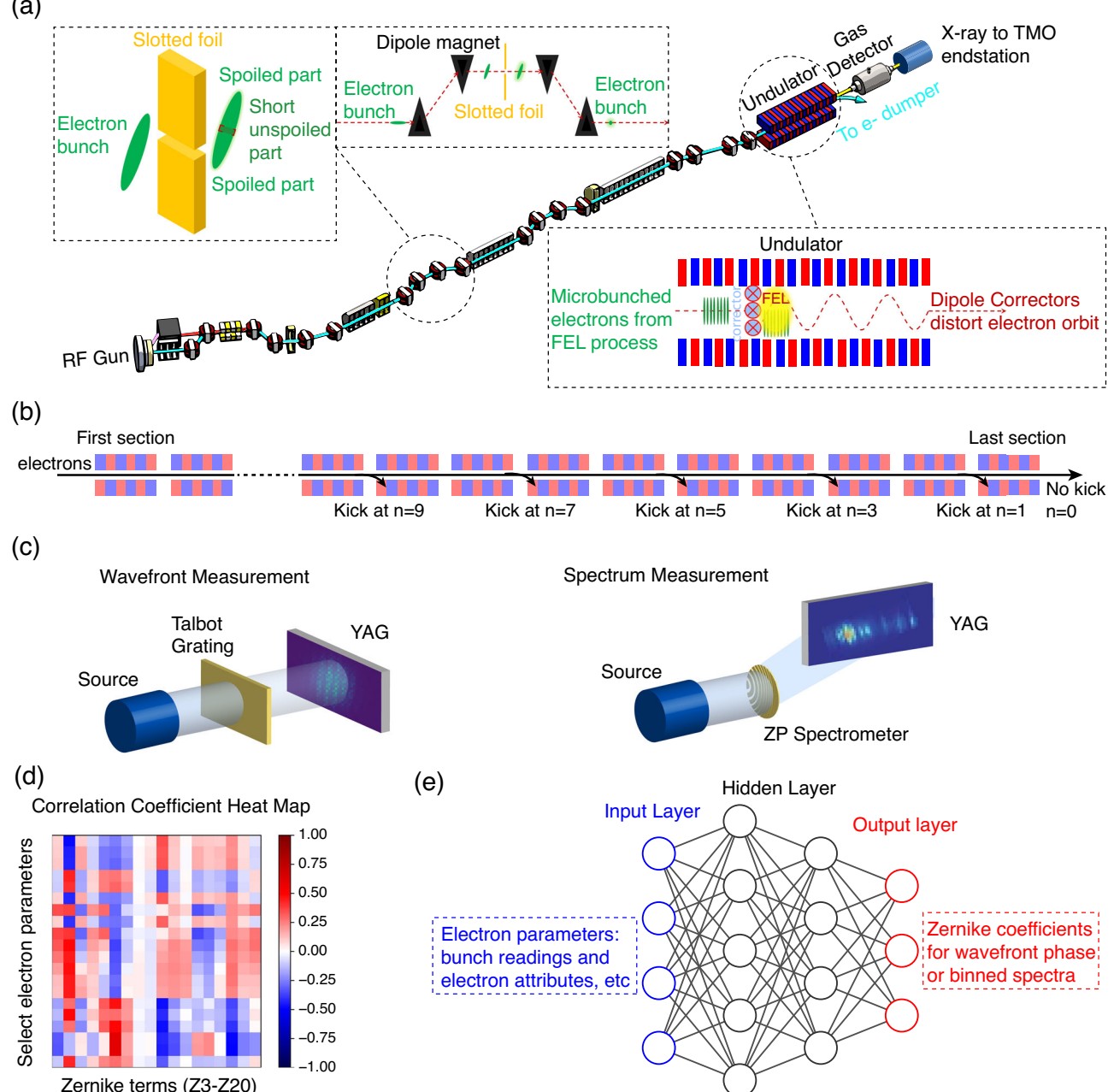

**Fig. 1 | Overview of experimental setup and data analysis. a** A schematic of the experiment in the detuning configuration with undulators and a slotted foil to produce short electron bunches. The X-ray pulses were delivered to the Time-resolved atomic, Molecular and Optical Science (TMO) instrument[31], where the X-ray diagnostics were located downstream of the instrument's Kirkpatrick-Baez (KB) focusing mirrors. **b** The electron bunch was kicked at various locations in the undulator, with $n$ sections before the final section, where $n = 0, 1, 3, 5,$ and 9, with 0 indicating no kicking. **c** Single-shot X-ray wavefronts were measured using a Talbot wavefront sensor[21-23]. The single-shot X-ray spectra were measured using an off-axis X-ray zone plate spectrometer[24, 25]. **d** From the recorded single-shot electron and X-ray data, a heat map displaying the Pearson correlation coefficients is produced, highlighting the relationship between the electron parameter inputs with the X-ray wavefront parameter outputs (Zernike coefficients of the X-ray wavefront phase). Each cell represents a correlation coefficient, with red indicative of a positive correlation and blue for a negative. The heat map shown is a subset of the data due to the large number recorded. **e** The electron and X-ray data were then used to train an artificial neural network (ANN). An illustrative diagram is shown representing a multilayer perceptron (MLP) model. The architecture consists of an input layer, several hidden layers, and an output layer, with the inputs being electron parameters and the outputs being X-ray beam properties. The diagram does not reflect the actual numbers for layers or nodes. The "Methods" section describes all parameters used.

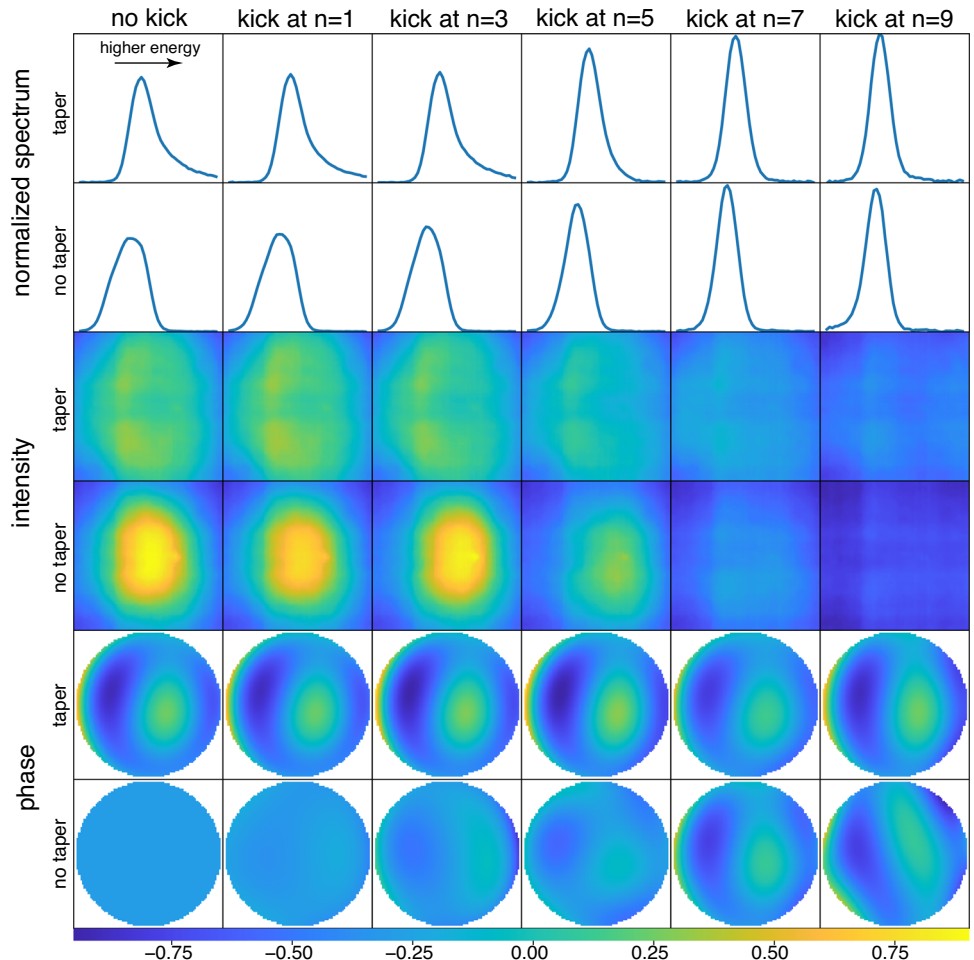

**Fig. 2 | X-ray spectra and wavefronts for different kicking locations and tapers.** The X-ray wavefront and spectrum were measured for various operating configurations, and each subplot displays the average measurement result of a particular case. The six columns correspond to six different kicking positions along the undulator, with $n = 0, 1, 3, 5, 7$, and 9 sections before the final undulator section, where 0 indicates no kicking. The six rows are grouped into three categories, displaying the normalized X-ray spectra, wavefront intensities, and phases, respectively. Within each category, the two rows show the results with and without undulator taper, respectively. The color bar at the bottom is only applicable to the phase maps. The wavefront phase plots have had the defocus and astigmatism terms removed and used the no taper no kick case as the reference to enhance the illustration of high-order phase differences. It is evident to see the differences in spectrum, wavefront intensity, and phase among cases. Please note that all plots are displayed in pixel units and are not calibrated to energy or length units due to experimental limitations during the run. For the purpose of this study, they are not required, but it would be desired for future studies.

results in low energy components in the spectra. The mean similarity between the predicted and measured spectra is 0.999 for average spectra, and 0.924 for single-shot spectra. Refer to the "Methods" section for further information on the prediction error and accuracy evaluation. With the single-shot measurements of a comprehensive collection of electron parameters, we can determine the overall spectrum of the X-ray beam delivered to the end station. The spectral resolution relies on the measurements obtained from the zone plate spectrometer as detailed in the "Methods" section on data acquisition. It is worth mentioning that the spikiness observed in a single-shot spectrum is a random occurrence and cannot be predicted due to the stochastic nature of XFEL startup and the inability to make measurements at the single-electron level. However, what holds significance is the envelope of the single-shot spectrum, as it provides information about the central frequency, bandwidth, and spectral tails at high energy for tapered cases and the tails at low energy for no taper cases. These distinctive features are illustrated in Fig. 4.

Furthermore, we also built and trained neural network models to perform classification tasks. We used either the wavefront phase Zernike coefficients or the electron parameters to predict the operation configuration from among the twelve options. The prediction

accuracy is remarkable, reaching 99% when given the electron parameters and 87% when given the wavefront phase Zernike coefficients at the single-shot level.

## Shot-to-shot variations

We utilized a similar technique to predict shot-to-shot variations in the single-shot X-ray wavefront phase within a single run using full SASE beams. Specifically, we employed a neural network to map electron parameter readings from the undulators to the measured single-shot X-ray wavefront phase. The results, depicted in Fig. 5a, illustrate the standard deviations of (1) the measured wavefront phase, (2) the predicted wavefront phase, and (3) the RMS prediction errors of the wavefront phase over all shots in the test dataset. The measured and predicted wavefront phases exhibit similar shot-to-shot variations, as evidenced by their comparable standard deviations for each Zernike term, particularly the two primary Zernike terms that contribute the most to shot-to-shot variations. The decrease in the variation of the difference between the measured and predicted wavefront phases indicates that the model has learned something that has reduced the difference to a level below shot-to-shot variations, and the remaining variation is likely due to shot-to-shot noise. Based on the single-shot wavefront phase data, the RMS

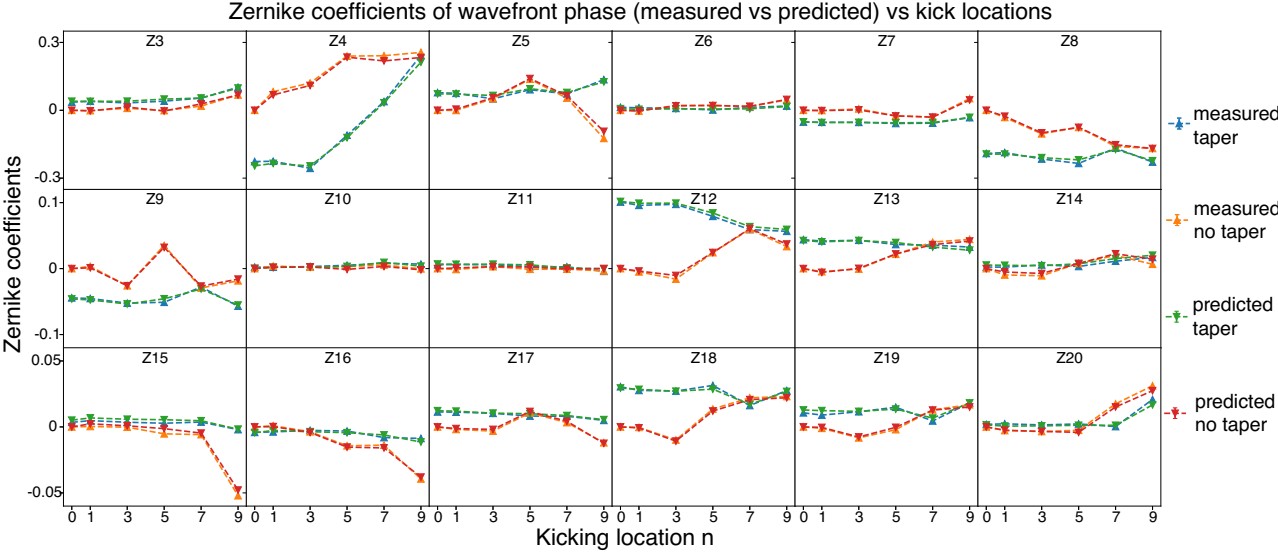

**Fig. 3 | Measured and predicted average Zernike coefficients for different kicking locations and tapers.** The Zernike coefficients obtained from both the measurements and predictions were averaged for each case, and each subplot displays a single Zernike term. The kicking locations are marked on the x-axis, with tapered cases shown in blue and green, and non-tapered cases in orange and red. The blue and orange lines indicate the measured Zernike coefficients, while the red and green lines represent the predicted Zernike coefficients from the trained model. The Zernike coefficients of the no kick no taper case (leftmost orange point in each subplot) are set to zero as a baseline for better illustration and comparison. The predicted and measured Zernike coefficients almost completely overlap with each other.

prediction error between the predicted and measured wavefront phase is determined to be 0.141 rad. Additionally, the standard deviation of the wavefront phase from shot to shot is calculated to be 0.269 rad. Consequently, the estimated relative error for predicting shot-to-shot fluctuations is ~52%. Refer to the "Methods" section on prediction error and accuracy evaluation for further details.

Figure 5b presents the measurement and prediction results from the test dataset, based on Zernike coefficients (Z3-Z8) versus an example electron beam parameter (electron x coordinate from a beam position monitor). It is worth noting that while a single electron parameter is depicted against Zernike coefficients in this figure, these coefficients are multivariate and rely on the complete set of electron parameters. The figure demonstrates how Zernike coefficients change as electron beam parameters vary and how the model's predictions compare to the measured data. Figure 5b indicates that the model has captured the correlations between Zernike coefficients and that selected single electron parameter, as well as the variation or dispersion among shots that arises from other electron parameters. The slight reduction in variation or dispersion from the prediction in Fig. 5b and the difference between measured and predicted wavefront phase in Fig. 5a may both be indications of noise sources (either systematic or measurement noise) that were not learned by the model.

Single-shot prediction is vital for XFEL X-ray imaging that relies on wavefront phase, as well as any other experiment that depends on X-ray intensity or spectra on the sample. This capability enables us to determine the wavefront phase in cases where direct, single-shot, in-situ wavefront measurements are not feasible, particularly for the exact shot pulse being used for single-shot imaging due to shot-to-shot variations of XFEL pulses. Although using a grating to split XFEL X-ray beams and measure the wavefront phase and spectrum to determine the X-ray delivered to experiments is possible, it would significantly increase the complexity of the experimental setup, consume more time and space, and result in a loss of photon flux.

## Discussion

Our recent experiments at LCLS have confirmed that ANN models can be trained on experiment data to accurately predict XFEL pulse properties such as wavefront and spectra using electron bunch parameters as inputs. The study aims to emphasize the valuable insights provided by electron diagnostics in predicting X-ray output. While acknowledging the complexity of XFEL physics, the study demonstrates the efficacy of the MLP model in capturing the nonlinear relationships between electron parameters and X-ray characteristics. This capability will simplify virtual diagnostics for single-shot X-ray pulses and facilitate electron diagnostics, optimization, and tuning to achieve optimal or desired X-ray output.

Optimal performance in ANN training and tuning necessitates a large dataset encompassing a diverse sample space. In this work, we utilized readily available shot-to-shot recorded electron beam parameters while measuring the XFEL beam, without investing additional effort in obtaining innovative electron measurements. However, to explore further avenues for improvement, it is worth considering to introduce additional parameters that provide a more comprehensive and in-depth characterization of electron information. By incorporating such parameters, the method presented here has the potential to enhance the model's robustness, reliability, and overall performance. For instance, the Convolutional Neural Network (CNN) can serve as a subnet for processing 2D electron parameters, specifically electron time-energy distribution images obtained from the X-band Transverse CAVity (XTCAV) diagnostic system[34]. By leveraging its ability to recognize learned patterns in these 2D inputs, the CNN can effectively extract relevant features. Moreover, to capture temporal pulse-pulse correlations, alternative models such as recursive neural network or transformer can be employed. These sequential models excel at extracting features related to the contextual information within the pulses, thereby providing a more comprehensive understanding of the data.

Similarly, further avenues for improvement can be made in the areas of X-ray diagnostics as well. Improvements in the performance of existing diagnostic tools as well as introduction of additional measurement capabilities in the future, for example the ability to measure temporal characteristic of the X-ray beam, can improve the overall performance of this type of model. Incorporation of the various instrument optics performance modeling and their optomechanic or other tuning parameters specific to each instrument can allow the integration of information from any X-ray optics induced characteristics or fluctuations in the beam prior to interaction with the sample.

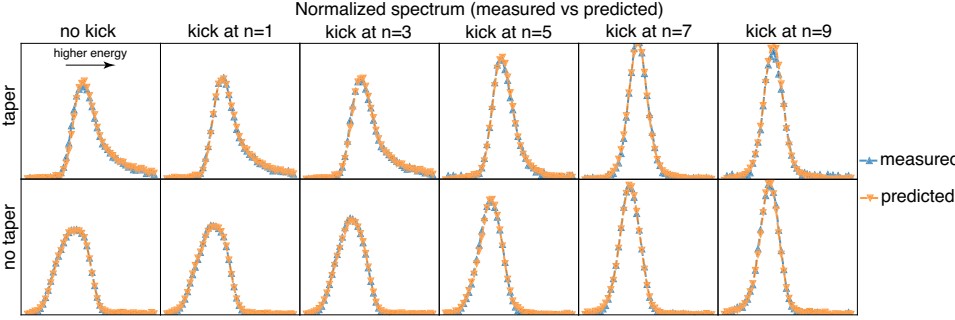

**Fig. 4 | Measured and predicted average X-ray spectra for different kicking locations and tapers.** The spectrum numbers obtained from both the measurements and predictions were averaged for each case, and each subplot displays a single case among the various operating configurations. The measured data is shown in blue, while the model predictions are in orange. The kicking locations are shown in different columns, and the taper and non-taper cases are displayed in the two rows, respectively. The predicted and measured spectra almost completely overlap with each other. The intensity of the spectrum data was normalized, so only the spectral shape was considered, without the intensity information. The spectra were plotted against pixels on the yttrium aluminum garnet (YAG) screen, without any calibration to energy units, as it was not necessary. A downstream kick resulted in a larger variation in the spectrum compared to an upstream kick. Moreover, the taper and non-taper cases exhibit opposite skewness in the spectrum.

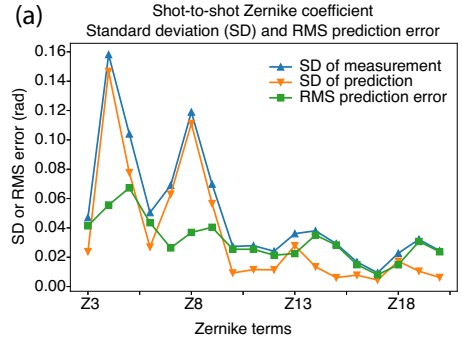

**Fig. 5 | Measured and predicted single-shot Zernike coefficients and shot-to-shot variations. a** The standard deviations (SD) of measurements, predictions, and RMS prediction error for the Zernike coefficients of all single shots from the test dataset are displayed for a single routine run with a full SASE beam. The measurement and prediction variations are similar, particularly for the two primary Zernike terms, Z4 and Z8. The smaller RMS prediction errors compared to the SD of measurement indicate that the model has accurately learned the systematic shot-to-shot variations. **b** The figure shows examples of measurements and predictions from the test dataset for Zernike coefficients (Z3-Z8) versus an example electron parameter (electron x-coordinate from a beam position monitor at a specific location within the Linac-To-Undulator Soft Line area, in units of standard deviation). The plot illustrates how the Zernike coefficients change as the electron parameters vary. The predictions not only capture the correlations between the Zernike coefficients and electron parameters shown but also demonstrate the variation or dispersion that results from other electron parameters.

This can lead to a higher fidelity predictive capability in the model as well as improved overall tuning of the accelerator and optics systems for an experiment.

## Methods
### The slotted electron beam configurations
To understand the mechanism behind the presence of high-order modes in XFEL pulses, we intentionally generate short electron bunches that resemble a single coherent spike. If we used a long electron bunch, it would result in many (order of 100) coherent spikes[4], which would be different transverse eigenmodes in the post-saturation regime of the XFEL. Observing these different modes becomes difficult when many spikes interfere with each other as they hit the wavefront sensor.

We utilized a slotted foil to spoil the majority of the electron bunch, leaving only a small, ultrashort portion[32], as shown in Fig. 1a. This ultrashort, unspoiled portion lases and generates an ultrashort XFEL pulse, which allows us to manipulate the electron bunch properties and undulator configuration to excite different high-order eigenmodes. Additionally, to effectively excite high-order modes, we perturb the electron orbit in the undulator by kicking it at specific locations, shown in Fig. 1b. The kicking occurs at $n$ sections before the final undulator section with $n = 0, 1, 3, 5, 7$, and 9, where 0 means no kicking.

In the high-gain XFEL, the slowly varying envelope function of the electric field has the form:

$$E = e^{-i\Omega\tau} e^{iq_\parallel \zeta} \psi(\mathbf{x}), \tag{1}$$

where the dimensionless variables measuring spatial and temporal variations are:

$$\tau = \omega_w t, \zeta = k_r(z - v_0 t), \mathbf{x} = \sqrt{2k_0 k_w}\mathbf{r}, \tag{2}$$

with $\mathbf{r}$ being the transverse coordinates, $z$ the longitudinal coordinate, $t$ the time, $v_0$ the electron bunch longitudinal velocity, $\omega_w = k_w c = (2\pi/\lambda_w)c$ and $\lambda_w$ being the undulator period, $c$ being the speed of light in vacuum, $k_r = k_0 + k_w = 2\pi/\lambda_0 + k_w$ and $\lambda_0$ being the radiation wavelength.

The eigenfrequencies $\Omega = \Omega_n(q_\parallel)$ and the eigenfunctions $\psi = \psi_n(q_\parallel, \mathbf{x})$ are determined by the dispersion relation[26]:

$$\left[\Omega - q_\parallel + \nabla_\perp^2 + \frac{\alpha}{\Omega^2}(\Omega - q_\parallel - 1)u(x)\right]\psi(\mathbf{x}) = 0, \tag{3}$$

where $\alpha = (n_0 \mu_0 e^4 A_w^2)/(2m^3 \gamma_0^3 \omega_w^2)$ with $n_0$ being the peak density of the electron bunch, $\gamma_0$, $e$, and $m$ being the Lorentz factor, the charge, and

the mass of the electron, respectively, $\mu_0$ being the vacuum permeability, and $A_w$ being the vector potential of the undulator.

It is now clear that to excite high-order eigenmodes $\psi_n(q_\parallel)$, the system should be detuned to support that particular eigenfrequency $\Omega_n(q_\parallel)$. In our experiment, we then introduced an energy chirp along the electron bunch to efficiently excite high-order modes.

Besides introducing energy chirp along the electron bunch for detuning, we can also adjust the taper of the undulator, since the XFEL wavelength is: $\lambda_{FEL} = \lambda_w(1 + K^2/2)/(2\gamma_0^2)$, tapering the undulator strength $K$ will directly detune $\lambda_{FEL}$. In the experiment, we study the evolution of high-order modes by setting the undulator sections in two states: no taper and optimal taper[33]. On top of these states, the undulator can be over-tapered to introduce the proper effective detuning for efficient excitation, guiding, and amplification of high-order eigenmodes.

## Data acquisition and preparation

The single shot data was recorded as two distinct datasets—one for the X-ray beam wavefront/spectra on the photon side and another for the electron parameter readings from undulators on the accelerator side. Both datasets recorded the single shot pulse energies, which were used to synchronize the two datasets on a single shot basis, thus ensuring that the X-ray and electron data is aligned for each individual shot.

The X-ray data includes the X-ray wavefront and spectrum. Highly accurate wavefront measurements were conducted using a Talbot wavefront sensor[35], which has recently been successfully demonstrated with XFEL radiation[21–23]. The number of Zernike terms required for an accurate representation of a wavefront phase depends on the complexity of the wavefront and the desired level of accuracy. To most XFEL experiments, the most important photon beam characteristics are focused beam position and profiles, which typically fluctuate shot-to-shot in current generation XFELs due to the SASE nature of lasing. Low order Zernike terms (up to Z15-Z21) can effectively capture the aberrations associated with those fluctuations, allowing a reasonably accurate determination on the beam features. In our specific case, considering both the absolute values and standard deviations of the higher-order Zernike coefficients to be very small compared to the dominant terms, we retrieved the wavefront phase and decomposed it into 21 Zernike coefficients (Z0-Z20) following the OSA/ANSI convention. By utilizing these Zernike coefficients, we were able to represent and characterize the wavefront phase. Each coefficient corresponds to a specific property of the wavefront, such as oblique and vertical astigmatism (Z3, Z5), defocus (Z4), trefoil (Z6, Z9), and coma (Z7, Z8). Decomposing the wavefront into Zernike polynomials serves as a featurization step, converting diverse forms of data into numerical representations suitable for basic machine learning algorithms.

For spectral measurements, we utilized an off-axis zone plate and captured the spectra on a YAG screen using a CCD camera. The spectrometer demonstrated a sub-eV spectral resolution (0.5–0.7 eV) in the vicinity of 530 eV. On the CCD, the pixel-to-eV ratio was 29 pixels per eV around 530 eV. To facilitate training and prediction, the resulting spectrum was subsequently binned into 50 values at a 6:1 ratio (equivalent to 0.2 eV per value after binning), offering a comprehensive representation of the overall spectrum shape.

We did not intentionally choose specific electron parameters and attributes; instead, we utilized all the directly accessible single-shot parameters. The model relied on a total of 192 parameters to generate the X-ray output. These electron parameters encompass readings from a range of sources such as bunch length monitors and beam position monitors at different sections (undulator soft line, linac-to-undulator soft line, electron dump soft line) and include electron beam positions (x and y coordinates), bunch charges, peak current, raw waveform, X-ray pulse energy, etc.

## Model training

To prepare the data for model training, we initially screened the pulse energy data to eliminate outliers by removing shots that were exceptionally weak or empty. In order to capture the intricate relationship between the electron beam parameters as input and X-ray output, we employed an MLP model. The MLP functions as a black box, taking the electron input and generating predictions for the corresponding X-ray output. Its focus is on establishing a nonlinear mapping rather than simulating the complex physics of XFEL systems.

The architecture of the MLP comprises several layers, including an input layer, three hidden layers with 256, 128, and 64 nodes respectively, and an output layer. The number of nodes in the input layer corresponds to 192 electron beam parameters, while the output layer consists of either 18 nodes for wavefront phase or 50 nodes for the spectrum. The electron parameters, which encompass parameters of the electron bunch and the undulators, serve as the input for the neural network. Prior to training, these parameters are normalized to enhance performance.

The output of the network is either the wavefront phase, represented by Zernike coefficients, or the normalized spectrum numbers. To ensure that the model accurately captures the nonlinear relationship and maintains generalization capability, we carefully select hyperparameters to prevent both underfitting and overfitting. The MLP utilizes the hyperbolic tangent (tanh) activation function, which allows for output normalization within the range of (−1, 1), effectively capturing both positive and negative influences from the input data.

For training the model, we employ the Mean Squared Error (MSE) as the loss function, along with dropout regularization (rate of 0.1) to prevent overfitting. An Adam optimizer and a batch size of 256 are utilized during the training process. We trained the model using 80% of approximately 10,000 total shots, while the remaining 20% was reserved for evaluating its predictive capabilities. To ensure the reliability of the model, we performed 5-fold cross-validation. This process involved dividing the data into 5 subsets and conducting training and evaluation on different combinations of these subsets. The consistently minimal errors observed during cross-validation indicated that the model was not prone to overfitting or selection bias.

## Prediction error and accuracy evaluation

When we have two 2D wavefront phase maps, the RMS difference between these wavefronts can be computed as $\sqrt{\|\Delta\mathbf{X}\|_2^2}$. Here, $\Delta\mathbf{X}$ represents the phase difference within the circular aperture. Alternatively, this difference can be expressed in terms of Zernike coefficients as $\|\Delta\mathbf{Z}\|_2 = \sqrt{\sum_j (\Delta Z_j)^2}$, where $\Delta Z_j$ signifies the discrepancy on each Zernike coefficient. This formula is used to calculate the RMS error between the measured wavefront and the predicted wavefront. Additionally, we can assess the shot-to-shot or case-to-case variations by considering $\Delta Z_j$ as the standard deviation of each Zernike coefficient. By dividing the RMS prediction error by the standard deviation of the wavefronts, the wavefront prediction error can be evaluated as a relative error.

To evaluate the accuracy of spectrum shape prediction, we measure the similarity between the predicted and measured spectra using the cosine similarity formula $S_C(\mathbf{A}, \mathbf{B}) = \frac{\mathbf{A} \cdot \mathbf{B}}{|\mathbf{A}|_2 |\mathbf{B}|_2}$. This calculation allows us to quantify the level of resemblance between the predicted and measured spectra, providing a metric for assessing the accuracy of the prediction.

## Data availability

The processed data subset can be accessed on Zenodo. Additional raw datasets that support the findings of this study are available from the corresponding authors upon request. Source data are provided with this paper.

## Code availability

The code used for the data analysis is available from the corresponding authors upon request.

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

## Acknowledgements

The authors thank Daniel Ratner for informative discussions about machine learning, and Franz-Josef Decker, Yuantao Ding, Andy Aquila, Matthieu Chollet and Peter Walter for experimental assistance and discussions. K. Li, Y. Liu, J. Wu, and A. Sakdinawat were supported by the U.S Department of Energy, Office of Science, Office of Basic Energy Sciences FWP No. 100622 at SLAC National Accelerator Laboratory, under contract No. DE-AC02-76SF00515. Use of the Linac Coherent Light Source (LCLS), SLAC National Accelerator Laboratory, is supported by the U.S. Department of Energy, Office of Science, Office of Basic Energy Sciences under Contract No. DE-AC02-76SF00515. Part of this work was performed at nano@stanford, supported by the National Science Foundation under award ECCS-2026822.

## Author contributions

A.S., J.W., K.L., and Y.L. conceived the idea, designed the experiments, and supervised the research. K.L. fabricated the X-ray diffractive optics for the X-ray diagnostic tools. M.L., X.C., M.S., K.L., and Y.L. carried out X-ray wavefront and spectral measurements. G.Z., J.W., A.A.L., H.S., and P.A.K. performed accelerator operations, electron parameters recording, and accelerator side data and model review. K.L. performed data preparation, data analysis, and model training. K.L., J.W., Y.L., and A.S. analyzed the results and wrote the manuscript. All authors commented on the manuscript.

## Competing interests

The authors declare no competing interests.
