## [Peer Review File · Nature Communications]

Prediction on X-ray Output of Free Electron Laser Based on Artificial Neural NetworksREVIEWER COMMENTS

Reviewer #1 (Remarks to the Author):

Overview and recommendations:

The authors present a neural network-based method for the analysis of x-ray output and free electron laser properties. The proposed method allows to map electron parameters obtained from undulators to measured x-ray output properties (waveform phases represented as Zernike coefficients) or energy spectrum. To highlight is the fact that the measurements for the neural network training and test data collection has been performed under different operational settings, which demonstrates the robustness of the proposed method against changes in operational modes. Another noteworthy advantage of presented method is that it opens the path to study the exact correlations between electron beam diagnostic and real-time x-ray output properties, which is crucial for precise FEL control and performance optimization.

The introduction includes a short but sufficient overview on previous related work, making clear the limitations of existing methods and how the x-ray output analysis can be improved by the proposed method. The results and methodology sections are presented in a comprehensive manner and include details on the data collection and analysis, only a few improvements should be added (please see the comments list below).

I recommend the manuscript for publication after the authors consider and implement/address the following comments:

- “Results” section: description of undulator sections and kicks locations is slightly misleading since Figure 1 does not show any location with $n = 0$.
- Figure 1: please improve the readability of the figure 1a), some of the text is not readable, also the elements in the layout are difficult to be recognised.
- Figure 1e) does the network have exactly 2 hidden layers? Since it was not mentioned in the “Model training” section it is not clear, if the representation corresponds to the used neural network or if it is a generic example of a feed-forward architecture. Same applied to the number of input and output nodes. I would suggest to specify the exact numbers in the figure.
- Section “Different configurations”: the prediction error is given in absolute units, I would suggest to report the relative error to present the model accuracy, which is easier to understand for the readers who are not familiar with the common values for predicted properties.

- Figure 5: please add the notation a) and b) to the subfigures.

Two paragraphs before the “Discussion” section: the authors discuss the indications of possible noise sources - this section would be more complete if the authors would include a discussion on the limitations of presented method and potential improvements, e.g. which other machine learning algorithms can be used in order to include modelling/ prediction of uncertainties/ noise level.

- Section “Model training”: the authors should add a few details on methodology for the choice of the neural network parameters selection (activation function, number of layers.) The collected data seems to be divided into training and test sets only, meaning that no cross-validation has been performed to verify the model parameters choice?

Reviewer #2 (Remarks to the Author):

The authors successfully estimated the characteristics of an XFEL by analyzing measurements of an electron beam. The estimated properties of XFEL can provide valuable information for experimental users and the XFEL can be fully utilized by using electron beam information only. Given that XFEL is a non-linear process and highly sensitive to various electron beam parameters, I believe that Artificial Neural Networks (ANN) represent one of the most effective methods for the inference of XFEL properties. However, numerous pioneering works on ANN and other machine-learning techniques in the field of particle accelerators have already been reported. I consider the application of ANN in this manuscript as one of the use cases. While this journal encourages new ideas and pioneering works, the approach presented in this manuscript appears to be similar to previous pioneering works. Since solving this kind of machine learning problem requires a large amount of high-quality training data, it would have been beneficial if the authors had introduced some novel aspects in the measurement of electron beams and XFELs. However, the measurement methods employed in this manuscript are conventional. Nevertheless, this work serves as a commendable example of estimating XFEL properties using ANN. There are several other journals dedicated to light sources or accelerators where this work could be better suited for publication. While this manuscript appears to be concise, I assume that the authors tailored it to fit the format of this particular journal. If this manuscript were submitted to another journal, I would recommend providing more detailed information, including the specific structure of the ANN, the selection process for electron beam parameters, XFEL simulations, and so on. Furthermore, it would be beneficial to include a physical discussion on how the properties of the electron beam impact the XFEL spectrum and wavefronts. Since ANN operates as a black box model, it would be intriguing to see a physical interpretation of the ANN behavior.

Reviewer #3 (Remarks to the Author):

The authors describe in their manuscript the application of a neural network to predict the properties of x-ray radiation produced by a free-electron laser. For a number of different configurations as well as for single-shot and averaged data predictions of x-ray wave front parameters and the spectral shape of the x-ray pulse are generated, which show a remarkable similarity to measured data for the identical configurations. This result is remarkable and could be an important milestone towards usage of artificial networks in operation large and complex accelerators.

While overall being a relevant result, the authors seem to stay too much at the surface of the problem and do not describe the requirements to such an analysis, e.g. which values are really important and with what kind of resolution. Neither they describe why they selected these specific properties and the ability of the method to become generalized and applied to other properties. While the achieved agreement of predicted and measured data is impressive it would be good to include a discussion why there is such good agreement and what are the conditions to achieve good agreement. Such a discussion would contribute enormously to understand the requirements to apply the method.

I consider the paper worth publications in Nature Communications, due to its novelty, the relevance to the overall field, and since the description is made accessible to the non-expert. I would nonetheless suggest to apply several improvements before accepting the paper for publication. Some are suggested above, other more specific to the written text are listed below.

Comments about content and editing:

- In several instances throughout the paper the authors stress the complexity to measure the x-ray wavefront and spectrum. However, not only the authors use the respective instrumentation, it is as well installed at many free electron lasers worldwide. The value of using artificial networks to predict x-ray properties is thus probably are different ones: this method potentially predicts x-ray properties hidden or not measurable and it offers a much more direct method to predict and optimize accelerator operation. The authors should consider to highlight these advantages more and deemphasize the issue of complexity in determining spectra and wavefront.

- While the almost superficial discussion of the considered x-ray properties brings the advantage of readability for the non-expert, it would be really interesting to understand the relation of the shown results relative to resolution. This becomes in particular obvious for the spectral properties, considering only very smooth spectra in stark contrast to real x-ray SASE FEL spectra. The authors should consider to include a paragraph explaining how their results depend on the relative resolution. There is no discussion of the single-shot predictions of spectra, which seems to indicate that the method is not applicable here.

- Similarly it would be important to understand how relevant the prediction of the 18 Zernicke coefficients is in order to predict wavefronts for an experiment – is the wavefront with this number highly overdetermined or still underdetermined. Which other factors play a role and why haven't these been considered ?
- The readability of the paper might increase if the introduction would include a part about the resolution and, as well, something about the physics dependence of the x-ray properties from the FEL settings and how or if these physics dependence has been considered or not.
- Figure 1 is very complex and has a long, rather descriptive caption. The authors should consider to simplify the figure and to make the caption more concise. As a suggestion one might consider to remove (d) and (e) and rather make these separate figures. (d) is anyway not really comprehensive.
- Figure 2 shows experimental averaged results of spectra, intensity and phase. Intensity and phase are later not considered, spectrum is shown again in figure 4. I suggest this figure to be moved to the method when describing the experimental results. The contrast could be better, in particular for the phase plots.
- On page 5 a very short paragraph describes the relation of Zernicke coefficients to wavefronts. This is very short and not comprehensible to the expert. Better extend and add to methods ? As well a transfer matrix is described (partially shown in Fig. 1), but nothing is said about the relevance or even uniqueness of this transformation.
- Figure 3 is impressive in that the lines almost not differ. However, it is very difficult to read for this very reason. The relevance of the finding, despite being an impressive match, remains unclear as it is not discussed how these coefficients describe the real wavefront.
- In figure the marking of part a. and b. needs to be added. In the caption it says: 'small values', but since nowhere absolute values are indicated it is fully unclear what small is. Either here a relative comparison needs to be made or the absolute value be compared to absolute values of the coefficients.
- On page 8 top it says ' The decrease in the variation of the ...', however it is not described in comparison to what, or function of what the values decrease. This would be important for the understanding of this part.
- Below the rms difference of the single-shot wavefront phase is quoted to be 0.141 rad, while two pages before the seemingly same value is given as 0.138 rad. Why is there a difference or are these different values ? This should be better explained.

Response to the reviewers

We appreciate the time and effort the reviewers have invested in evaluating our work. We have carefully considered their comments and suggestions and have made the necessary revisions to address their concerns. Our point-by-point responses to each of the reviewers' comments are shown in blue and revisions to our manuscript are shown in red below, and highlighted in the manuscript.

The abstract and figure captions were revised to meet the word limit and formatting requirements.

Reviewer #1 (Remarks to the Author):

Overview and recommendations:

The authors present a neural network-based method for the analysis of x-ray output and free electron laser properties. The proposed method allows to map electron parameters obtained from undulators to measured x-ray output properties (waveform phases represented as Zernike coefficients) or energy spectrum. To highlight is the fact that the measurements for the neural network training and test data collection has been performed under different operational settings, which demonstrates the robustness of the proposed method against changes in operational modes. Another noteworthy advantage of presented method is that it opens the path to study the exact correlations between electron beam diagnostic and real-time x-ray output properties, which is crucial for precise FEL control and performance optimization.

The introduction includes a short but sufficient overview on previous related work, making clear the limitations of existing methods and how the x-ray output analysis can be improved by the proposed method. The results and methodology sections are presented in a comprehensive manner and include details on the data collection and analysis, only a few improvements should be added (please see the comments list below).

I recommend the manuscript for publication after the authors consider and implement/address the following comments:

- “Results” section: description of undulator sections and kicks locations is slightly misleading since Figure 1 does not show any location with $n = 0$.

We added the $n=0$ (no kicking) annotation to Fig. 1(b).

- Figure 1: please improve the readability of the figure 1a), some of the text is not readable, also the elements in the layout are difficult to be recognised.

We have vectorized the entire figure and made it clearer, improving its readability.

- Figure 1e) does the network have exactly 2 hidden layers? Since it was not mentioned in the “Model training” section it is not clear, if the representation corresponds to the used

neural network or if it is a generic example of a feed-forward architecture. Same applied to the number of input and output nodes. I would suggest to specify the exact numbers in the figure.

Fig. 1(e) is an illustrative diagram representing a multilayer perceptron (MLP) model. To clarify, we added the following sentences to the caption, and included more details in the Method - Model training section. The diagram does not reflect the actual numbers for layers or nodes. The Methods section describes all parameters used.

- Section “Different configurations”: the prediction error is given in absolute units, I would suggest to report the relative error to present the model accuracy, which is easier to understand for the readers who are not familiar with the common values for predicted properties.

We have added a subsection titled “Prediction error and accuracy evaluation” in the Method section. This subsection provides detailed information on how absolute and relative errors were calculated for both the wavefront and spectrum. Additionally, we report the specific numerical values in the Result section.

- Figure 5: please add the notation a) and b) to the subfigures.

We added the notation (a) and (b) to Fig. 5.

Two paragraphs before the “Discussion” section: the authors discuss the indications of possible noise sources - this section would be more complete if the authors would include a discussion on the limitations of presented method and potential improvements, e.g. which other machine learning algorithms can be used in order to include modelling/ prediction of uncertainties/ noise level.

We have added the following paragraphs in the Discussion section to discuss current limitations and potential improvements:

Optimal performance in ANN training and tuning necessitates a large dataset encompassing a diverse sample space. In this work, we utilized readily available shot-to-shot recorded electron beam parameters while measuring the XFEL beam, without investing additional effort in obtaining novel electron measurements. To explore further avenues for improvement, it is worth considering to introduce new parameters that provide a more comprehensive and in-depth characterization of electron information. By incorporating such parameters, the method presented here has the potential to enhance the model’s robustness, reliability, and overall performance. For instance, the Convolutional Neural Network (CNN) can serve as a subnet for processing 2D electron parameters, specifically electron time-energy distribution images obtained from the X-band Transverse CAVity (XTCAV) diagnostic system [34]. By leveraging its ability to recognize learned patterns in these 2D inputs, the CNN can effectively extract relevant features. Moreover, to capture temporal pulse-pulse correlations, alternative models such as recursive neural network or transformer can be employed. These sequential models excel at extracting features related to the contextual information within the pulses, thereby providing a more comprehensive understanding of the data.

Similarly, further avenues for improvement can be made in the areas of x-ray diagnostics as well. Improvements in the performance of existing diagnostic tools as well as introduction of new measurement capabilities in the future, for example the ability to measure temporal characteristic of the x-ray beam, can improve the overall performance of this type of model. Incorporation of the various instrument optics performance modeling and their optomechanic or other tuning parameters specific to each instrument can allow the integration of information from any x-ray optics induced characteristics or fluctuations in the beam prior to interaction with the sample. This can lead to a higher fidelity predictive capability in the model as well as improved overall tuning of the accelerator and optics systems for an experiment.

- Section “Model training”: the authors should add a few details on methodology for the choice of the neural network parameters selection (activation function, number of layers.) The collected data seems to be divided into training and test sets only, meaning that no cross-validation has been performed to verify the model parameters choice?

We have added more details in the Method - Model training section, discussing the specifics of the MLP architecture used in this paper:

To prepare the data for model training, we initially screened the pulse energy data to eliminate outliers by removing shots that were exceptionally weak or empty. In order to capture the intricate relationship between the electron input and x-ray output, we employed an MLP model. The MLP functions as a black box, taking the electron beam parameters as input and generating predictions for the corresponding x-ray output. Its focus is on establishing a nonlinear mapping rather than simulating the complex physics of XFEL systems.

The architecture of the MLP comprises several layers, including an input layer, three hidden layers with 256, 128, and 64 nodes respectively, and an output layer. The number of nodes in the input layer corresponds to 192 electron beam parameters, while the output layer consists of either 18 nodes for wavefront phase or 50 nodes for the spectrum. The electron parameters, which encompass parameters of the electron bunch and the undulators, serve as the input for the neural network. Prior to training, these parameters are normalized to enhance performance.

The output of the network is either the wavefront phase, represented by Zernike coefficients, or the normalized spectrum numbers. To ensure that the model accurately captures the nonlinear relationship and maintains generalization capability, we carefully select hyperparameters to prevent both underfitting and overfitting. The MLP utilizes the hyperbolic tangent (tanh) activation function, which allows for output normalization within the range of (-1, 1), effectively capturing both positive and negative influences from the input data.

For training the model, we employ the Mean Squared Error (MSE) as the loss function, along with dropout regularization (rate of 0.1) to prevent overfitting. An Adam optimizer and a batch size of 256 are utilized during the training process. We trained the model using 80%

of approximately 10,000 total shots, while the remaining 20% was reserved for evaluating its predictive capabilities. To ensure the reliability of the model, we performed 5-fold cross-validation. This process involved dividing the data into 5 subsets and conducting training and evaluation on different combinations of these subsets. The consistently minimal errors observed during cross-validation indicated that the model was not prone to overfitting or selection bias.

Reviewer #2 (Remarks to the Author):

The authors successfully estimated the characteristics of an XFEL by analyzing measurements of an electron beam. The estimated properties of XFEL can provide valuable information for experimental users and the XFEL can be fully utilized by using electron beam information only. Given that XFEL is a non-linear process and highly sensitive to various electron beam parameters, I believe that Artificial Neural Networks (ANN) represent one of the most effective methods for the inference of XFEL properties.

However, numerous pioneering works on ANN and other machine-learning techniques in the field of particle accelerators have already been reported. I consider the application of ANN in this manuscript as one of the use cases. While this journal encourages new ideas and pioneering works, the approach presented in this manuscript appears to be similar to previous pioneering works. Since solving this kind of machine learning problem requires a large amount of high-quality training data, it would have been beneficial if the authors had introduced some novel aspects in the measurement of electron beams and XFELs. However, the measurement methods employed in this manuscript are conventional.

Previous studies utilizing ANN in the context of XFEL have predominantly focused on electron-related tasks such as accelerator and undulator tuning. Our research takes a crucial step forward by establishing a connection between single-shot electron parameters and the resulting x-ray wavefront and spectrum output together on a single-shot basis. By doing so, it effectively demonstrates the valuable and informative nature of electron attributes in shaping XFEL outcomes. This is one of the most novel breakthroughs demonstrated in our work. We have the following sentences in the Introduction section.

To overcome these challenges, we developed a virtual diagnostics model based on artificial neural networks (ANNs) and shot-to-shot measurement data of both electron and x-ray beam parameters. Artificial neural networks (ANNs) are powerful tools for modeling complex nonlinear relationships, and exploration of their utility to overcome the limitations of conventional methods for accelerator optimization, tuning, and modeling is underway [14, 15, 16, 17]. The majority of machine learning models for XFELs have primarily focused only on the electron beam for tasks such as accelerator and undulator tuning and optimization [18, 19], with one study incorporating x-ray spectrometer data [20]. These studies were made possible due to the single-shot diagnostics of the electron beam implemented in the XFEL. With the recent development of high-accuracy single-shot x-ray wavefront sensors for both soft and hard x-rays at XFELs [21, 22, 23] and the development of single-shot

soft x-ray spectrometers based on off-axis zone plates for spectrum measurements [24, 25], x-ray properties can now be characterized routinely. These diagnostic tools enable us to measure the spatial amplitude and phase, as well as the spectral qualities of the x-ray beam and allow us to further combine the x-ray diagnostics data with that of the electron beam diagnostics data into a model based on ANNs.

We then emphasized the arguably most crucial parameters in delivering high-quality coherent XFEL pulses, which are detuning and tapering. These parameters manifest themselves in exciting high-order modes, which can degrade the spatial coherence of the XFEL pulses. However, on the other hand, some scientific users might desire certain types of high-order modes for their research purposes. In this work, both the spatial and spectral properties are measured using a wavefront sensor and a spectrometer, allowing for the extraction of correlations that guide the optimization of the XFEL pulse. Indeed, by properly adjusting these two parameters, successful demonstration of a donut-shaped high-order mode is achieved, which represents one of the novel findings in this study.

Detuning plays a critical role in determining XFEL modes through the dispersion relation equation [2, 3, 26, 27, 28, 29] and can excite high-order modes [30]. In conjunction with tapering of the undulators, amplification of these high-order modes are expected. Both the routine case and the specialized cases of detuning, tapering, and kicking were chosen to demonstrate and understand the utility and limitations of the ANN-based virtual diagnostics model.

Nevertheless, this work serves as a commendable example of estimating XFEL properties using ANN. There are several other journals dedicated to light sources or accelerators where this work could be better suited for publication. While this manuscript appears to be concise, I assume that the authors tailored it to fit the format of this particular journal. If this manuscript were submitted to another journal, I would recommend providing more detailed information, including the specific structure of the ANN, the selection process for electron beam parameters, XFEL simulations, and so on.

We have added additional texts and paragraphs in the Method - Model training section, addressing detailed ANN structure and the selection process for electron beam parameters, etc.

To prepare the data for model training, we initially screened the pulse energy data to eliminate outliers by removing shots that were exceptionally weak or empty. In order to capture the intricate relationship between the electron input and x-ray output, we employed an MLP model. The MLP functions as a black box, taking the electron beam parameters as input and generating predictions for the corresponding x-ray output. Its focus is on establishing a nonlinear mapping rather than simulating the complex physics of XFEL systems.

The architecture of the MLP comprises several layers, including an input layer, three hidden layers with 256, 128, and 64 nodes respectively, and an output layer. The number of nodes

in the input layer corresponds to 192 electron beam parameters, while the output layer consists of either 18 nodes for wavefront phase or 50 nodes for the spectrum. The electron parameters, which encompass parameters of the electron bunch and the undulators, serve as the input for the neural network. Prior to training, these parameters are normalized to enhance performance.

The output of the network is either the wavefront phase, represented by Zernike coefficients, or the normalized spectrum numbers. To ensure that the model accurately captures the nonlinear relationship and maintains generalization capability, we carefully select hyper-parameters to prevent both underfitting and overfitting. The MLP utilizes the hyperbolic tangent (tanh) activation function, which allows for output normalization within the range of (-1, 1), effectively capturing both positive and negative influences from the input data.

We did not intentionally choose specific electron parameters and attributes; instead, we utilized all the directly accessible single-shot parameters. The model relied on a total of 192 parameters to generate the x-ray output. These electron parameters encompass readings from a range of sources such as bunch length monitors and beam position monitors at different sections (undulator soft line, linac-to-undulator soft line, electron dump soft line) and include electron beam positions (x and y coordinates), bunch charges, peak current, raw waveform, x-ray pulse energy, etc.

We have incorporated additional references (highlighted in the manuscript) to offer more comprehensive information on the electron bunch parameters, undulator settings, and XFEL properties. These additions aim to provide detailed support for the motivations behind our research.

We have also included additional texts to provide further explanations for the observed seemingly counterintuitive results. These results highlight the powerful scheme devised, designed, and demonstrated in this work, which facilitates the ability to utilize detuning for either the generation or suppression of high-order modes.

Indeed, the experiments revealed interesting XFEL physics when certain parameters of the electron bunch and the undulators are varied. In the no-taper case, due to the fact that the electrons are continuously losing energy, the radiation spectrum is skewed toward the red-shift side. For the taper case, the taper was over-tapered to introduce a detuning to set the resonant frequency in the blue-shift side compared to the radiation frequency in the exponential growth region, i.e., before the tapered region. Thus, the microbunching will now support high-order modes according to the dispersion relation discussed below in the Method Section. In our case, the donut mode is excited, as shown in the intensity plot in Fig. 2, while the spectrum shows spectral tails at high energy, as seen in the normalized spectrum plot in Fig. 2.

Further texts are integrated with the physical discussion answering the question below and lay down the foundation of the theory behind this work.

Furthermore, it would be beneficial to include a physical discussion on how the properties of the electron beam impact the XFEL spectrum and wavefronts. Since ANN operates as a black box model, it would be intriguing to see a physical interpretation of the ANN behavior.

We have added a few paragraphs to include a physical discussion on how the properties of the electron beam impact the XFEL spectrum and wavefronts. In particular, we emphasize the importance of invoking detuning as an effective knob to suppress or excite high-order modes.

In the high-gain XFEL, the slowly varying envelope function of the electric field has the form:

$$E = e^{-i\Omega\tau} e^{iq_{\parallel}\zeta} \psi(\mathbf{x}),$$

where the dimensionless variables measuring spatial and temporal variations are:

$$\tau = \omega_w t, \quad \zeta = k_r(z - v_0 t), \quad \mathbf{x} = \sqrt{2k_0 k_w} \mathbf{r},$$

with \mathbf{r} being the transverse coordinates, z the longitudinal coordinate, t the time, v_0 the electron bunch longitudinal velocity, $\omega_w = k_w c = (2\pi/\lambda_w)c$ and λ_w being the undulator period, c being the speed of light in vacuum, $k_r = k_0 + k_w = 2\pi/\lambda_0 + k_w$ and λ_0 being the radiation wavelength.

The eigenfrequencies $\Omega = \Omega_n(q_{\parallel})$ and the eigenfunctions $\psi = \psi_n(q_{\parallel}, \mathbf{x})$ are determined by the dispersion relation [26]:

$$\left[\Omega - q_{\parallel} + \nabla_{\perp}^2 + \frac{\alpha}{\Omega^2} (\Omega - q_{\parallel} - 1) u(x) \right] \psi(\mathbf{x}) = 0,$$

where $\alpha = (n_0 \mu_0 e^4 A_w^2) / (2m^3 \gamma_0^3 \omega_w^2)$ with n_0 being the peak density of the electron bunch, γ_0 , e , and m being the Lorentz factor, the charge, and the mass of the electron, respectively, μ_0 being the vacuum permeability, and A_w being the vector potential of the undulator.

It is now clear that to excite high-order eigenmodes $\psi_n(q_{\parallel})$, the system should be detuned to support that particular eigenfrequency $\Omega_n(q_{\parallel})$. In our experiment, we then introduced an energy chirp along the electron bunch to efficiently excite high-order modes.

Besides introducing energy chirp along the electron bunch for detuning, we can also adjust the taper of the undulator, since the XFEL wavelength is: $\lambda_{\text{FEL}} = \lambda_w (1 + K^2/2) / (2\gamma_0^2)$, tapering the undulator strength K will directly detune λ_{FEL} . In the experiment, we study the evolution of high-order modes by setting the undulator sections in two states: no taper and optimal taper [33]. On top of these states, the undulator can be over-tapered to introduce the proper effective detuning for efficient excitation, guiding, and amplification of high-order eigenmodes.

Reviewer #3 (Remarks to the Author):

The authors describe in their manuscript the application of a neural network to predict the properties of x-ray radiation produced by a free-electron laser. For a number of different

configurations as well as for single-shot and averaged data predictions of x-ray wave front parameters and the spectral shape of the x-ray pulse are generated, which show a remarkable similarity to measured data for the identical configurations. This result is remarkable and could be an important milestone towards usage of artificial networks in operation large and complex accelerators.

While overall being a relevant result, the authors seem to stay too much at the surface of the problem and do not describe the requirements to such an analysis, e.g. which values are really important and with what kind of resolution. Neither they describe why they selected these specific properties and the ability of the method to become generalized and applied to other properties. While the achieved agreement of predicted and measured data is impressive it would be good to include a discussion why there is such good agreement and what are the conditions to achieve good agreement. Such a discussion would contribute enormously to understand the requirements to apply the method.

While all electron parameters contribute to the x-ray output, the correlation coefficients indicate that variations in downstream monitors' electron parameters play a more significant role in the final x-ray output. We have included the following paragraph in the Method - Data acquisition and preparation section, providing details about the selection and attributes of the electron parameters:

We did not intentionally choose specific electron parameters and attributes; instead, we utilized all the directly accessible single-shot parameters. The model relied on a total of 192 parameters to generate the x-ray output. These electron parameters encompass readings from a range of sources such as bunch length monitors and beam position monitors at different sections (undulator soft line, linac-to-undulator soft line, electron dump soft line) and include electron beam positions (x and y coordinates), bunch charges, peak current, raw waveform, x-ray pulse energy, etc.

The good agreement observed in the figures is due to the fact that they represent comparisons of the averages. The root mean square (RMS) prediction errors on the averages are smaller than the errors in the single-shot measurements. The specific values are specified in the main body text. With a comprehensive set of electron parameters and attributes describing the behavior of the electron bunch, a well-trained model is expected to provide x-ray output that closely aligns with the measurements.

Detuning and tapering are arguably the most crucial aspects in achieving on-demand FEL x-ray pulse delivery. These fundamental factors drive the setup of different configurations to train the machine learning models. With the trained models, users can request specific XFEL pulses by assisting in setting up the electron bunch and undulator parameters. We have demonstrated that machine learning faithfully captures the subtle physics originating from detuning and tapering for the different configurations, as described in the manuscript. In summary, for the various configurations, detuning the electron bunch enables the excitation of high-order modes, while the taper and deliberately perturbed orbits amplify these

excited high-order modes. Our intention is to convince readers that other relevant parameters, such as pointing or chirp, should be equally controllable and expressible, as we have achieved success with the most essential physics parameters: detuning and tapering.

We added motivation and demonstrated the crucial reasons why we chose to work on detuning and tapering.

We modulate the electron beam parameters via different accelerator operational configurations in the XFEL, including both that of routine operations with full normal electron beams and exploration of the effect of detuning, tapering, and kicking of slotted electron beams, record the electron and x-ray beam parameters on a single-shot basis, and then train an ANN-based model using the data. Detuning plays a critical role in determining XFEL modes through the dispersion relation equation [2, 3, 26, 27, 28, 29] and can excite high-order modes [30]. In conjunction with tapering of the undulators, amplification of these high-order modes are expected. Both the routine case and the specialized cases of detuning, tapering, and kicking were chosen to demonstrate and understand the utility and limitations of the ANN-based virtual diagnostics model.

We have also added more references (highlighted in the manuscript) to support our statement regarding the importance of studying high-order modes in our novel work. With the inclusion of these references, we aim to clarify why we have chosen detuning and tapering for different configurations in model training. Our goal is to assist operators in setting up the electron bunch and undulator parameters to deliver XFEL pulses as requested by users.

Furthermore, we have provided a concise description of the overall physics in the Method section. The derived dispersion relation summarizes how specific electron properties impact x-ray pulse properties.

In the high-gain XFEL, the slowly varying envelope function of the electric field has the form:

$$E = e^{-i\Omega\tau} e^{iq_{\parallel}\zeta} \psi(\mathbf{x}),$$

where the dimensionless variables measuring spatial and temporal variations are:

$$\tau = \omega_w t, \quad \zeta = k_r(z - v_0 t), \quad \mathbf{x} = \sqrt{2k_0 k_w} \mathbf{r},$$

with \mathbf{r} being the transverse coordinates, z the longitudinal coordinate, t the time, v_0 the electron bunch longitudinal velocity, $\omega_w = k_w c = (2\pi/\lambda_w)c$ and λ_w being the undulator period, c being the speed of light in vacuum, $k_r = k_0 + k_w = 2\pi/\lambda_0 + k_w$ and λ_0 being the radiation wavelength.

The eigenfrequencies $\Omega = \Omega_n(q_{\parallel})$ and the eigenfunctions $\psi = \psi_n(q_{\parallel}, \mathbf{x})$ are determined by the dispersion relation [26]:

$$\left[\Omega - q_{\parallel} + \nabla_{\perp}^2 + \frac{\alpha}{\Omega^2} (\Omega - q_{\parallel} - 1) u(x) \right] \psi(\mathbf{x}) = 0,$$

where $\alpha = (n_0\mu_0e^4A_w^2)/(2m^3\gamma_0^3\omega_w^2)$ with n_0 being the peak density of the electron bunch, γ_0 , e , and m being the Lorentz factor, the charge, and the mass of the electron, respectively, μ_0 being the vacuum permeability, and A_w being the vector potential of the undulator.

It is now clear that to excite high-order eigenmodes $\psi_n(q_{\parallel})$, the system should be detuned to support that particular eigenfrequency $\Omega_n(q_{\parallel})$. In our experiment, we then introduced an energy chirp along the electron bunch to efficiently excite high-order modes.

Besides introducing energy chirp along the electron bunch for detuning, we can also adjust the taper of the undulator, since the XFEL wavelength is: $\lambda_{\text{FEL}} = \lambda_w(1 + K^2/2)/(2\gamma_0^2)$, tapering the undulator strength K will directly detune λ_{FEL} . In the experiment, we study the evolution of high-order modes by setting the undulator sections in two states: no taper and optimal taper [33]. On top of these states, the undulator can be over-tapered to introduce the proper effective detuning for efficient excitation, guiding, and amplification of high-order eigenmodes.

I consider the paper worth publications in Nature Communications, due to its novelty, the relevance to the overall field, and since the description is made accessible to the non-expert. I would nonetheless suggest to apply several improvements before accepting the paper for publication. Some are suggested above, other more specific to the written text are listed below.

Comments about content and editing:

- In several instances throughout the paper the authors stress the complexity to measure the x-ray wavefront and spectrum. However, not only the authors use the respective instrumentation, it is as well installed at many free electron lasers worldwide. The value of using artificial networks to predict x-ray properties is thus probably are different ones: this method potentially predicts x-ray properties hidden or not measurable and it offers a much more direct method to predict and optimize accelerator operation. The authors should consider to highlight these advantages more and deemphasize the issue of complexity in determining spectra and wavefront.

The sentences in the Introduction section indeed can be improved following the reviewer's suggestion. We have added the following:

First, the wavefront and spectrum of the XFEL pulse can be determined computationally, though this is a challenging task due to the complexity of the underlying physics, discrepancies between real-world and computational models, and the multitude of variables and parameters involved especially with the newer generation XFELs. Second, real-time non-destructive measurements of the energy spectral and spatial wavefront properties of the XFEL pulse delivered to a sample could also be implemented. One method to do this involves splitting the x-ray pulse into reference and experimental beams using a beam splitter and taking measurements on both beams from shot to shot. This, however, can increase experimental complexity, require additional instrumentation, which may not be feasible de-

pending on the physical constraints of the experimental setups, and reduces photon flux. In addition, accuracy would be highly determined by the quality and performance of the x-ray beam splitter optic.

With the recent development of high-accuracy single-shot x-ray wavefront sensors for both soft and hard x-rays at XFELs [21, 22, 23] and the development of single-shot soft x-ray spectrometers based on off-axis zone plates for spectrum measurements [24, 25], x-ray properties can now be characterized routinely. These diagnostic tools enable us to measure the spatial amplitude and phase, as well as the spectral qualities of the x-ray beam and allow us to further combine the x-ray diagnostics data with that of the electron beam diagnostics data into a model based on ANNs.

Nondestructively determining the wavefront and spectra of the exact single x-ray pulse delivered to a sample presents challenges due to the limited geometric and optical constraints of XFEL endstations. The process requires additional space and complexity in the endstation setup to overcome these constraints. Specialized optics and intricate measurement techniques must be accommodated, posing difficulties in accurately capturing the wavefront and spectra of the specific x-ray pulse being used.

In this context, the utilization of artificial neural networks (ANN) to predict x-ray properties offers a significant advantage. It enables real-time, non-destructive measurement of the x-ray pulse delivered to the sample or experiment. Traditional measurement methods, such as beam splitting, often result in a reduction in photon flux, making it challenging to precisely measure the desired x-ray pulse. Conversely, employing ANN-based prediction allows for estimating x-ray properties without compromising the photon flux, thereby facilitating accurate monitoring and real-time analysis of the utilized x-ray pulse.

- While the almost superficial discussion of the considered x-ray properties brings the advantage of readability for the non-expert, it would be really interesting to understand the relation of the shown results relative to resolution. This becomes in particular obvious for the spectral properties, considering only very smooth spectra in stark contrast to real x-ray SASE FEL spectra. The authors should consider to include a paragraph explaining how their results depend on the relative resolution. There is no discussion of the single-shot predictions of spectra, which seems to indicate that the method is not applicable here.

The smooth spectra presented in the figure represent the averaged spectra, while the spikiness is observed in the single-shot spectrum. Predictions are all single shots, and the averages are calculated based on the predicted single shots. When evaluating the prediction performance (comparing prediction vs measurement), spectral resolution is not a suitable metric, as both spectra have the same number of discrete values along the energy axis. To assess the relative error or discrepancy in spectral shape prediction, we have introduced a similarity metric, as described in the Method - Prediction error and accuracy evaluation section, and included the results in the Results section.

The mean similarity between the predicted and measured spectra is 0.999 for average spec-

tra, and 0.924 for single-shot spectra. Refer to the Method section for further information on the prediction error and accuracy evaluation.

To evaluate the accuracy of spectrum shape prediction, we measure the similarity between the predicted and measured spectra using the cosine similarity formula $S_C(\mathbf{A}, \mathbf{B}) = \frac{\mathbf{A} \cdot \mathbf{B}}{\|\mathbf{A}\|_2 \|\mathbf{B}\|_2}$. This calculation allows us to quantify the level of resemblance between the predicted and measured spectra, providing a metric for assessing the accuracy of the prediction.

It is worth noting that the spectral resolution relies on the measurements obtained from the zone plate spectrometer. For additional information regarding the specifications of the spectrometer, a dedicated paragraph has been included in the Method - Data acquisition and preparation section.

For spectrum measurement, we utilized an off-axis zone plate and captured the spectra on a YAG screen using a CCD camera. The spectrometer demonstrated a sub-eV spectral resolution (0.5-0.7 eV) in the vicinity of 530 eV. On the CCD, the pixel-to-eV ratio was 29 pixels per eV around 530 eV. To facilitate training and prediction, the resulting spectrum was subsequently binned into 50 values at a 6:1 ratio (equivalent to 0.2 eV per value after binning), offering a comprehensive representation of the overall spectrum shape.

We have included comments on the spikiness observed in single-shot spectra, emphasizing that the predictability lies in the envelope rather than the spikiness itself.

It is worth mentioning that the spikiness observed in a single-shot spectrum is a random occurrence and cannot be predicted due to the stochastic nature of XFEL startup and the inability to make measurements at the single-electron level. However, what holds significance is the envelope of the single-shot spectrum, as it provides information about the central frequency, bandwidth, and spectral tails at high energy for tapered cases and the tails at low energy for no taper cases. These distinctive features are illustrated in Fig. 4.

- Similarly it would be important to understand how relevant the prediction of the 18 Zernicke coefficients is in order to predict wavefronts for an experiment – is the wavefront with this number highly overdetermined or still underdetermined. Which other factors play a role and why haven't these been considered?

We have added a paragraph in the Method - Data acquisition and preparation section to explain why and how we decomposed the wavefront phase into the Zernike polynomial basis:

The number of Zernike terms required for an accurate representation of a wavefront phase depends on the complexity of the wavefront and the desired level of accuracy. To most XFEL experiments, the most important photon beam characteristics are focused beam position and profiles, which typically fluctuate shot-to-shot in current generation XFELs due to the SASE nature of lasing. Low order Zernike terms (up to Z15-Z21) can effectively capture the aberrations associated with those fluctuations, allowing a reasonably accurate determinations on the beam features. In our specific case, considering both the absolute values and standard deviations of the higher-order Zernike coefficients to be very small

compared to the dominant terms, we retrieved the wavefront phase and decomposed it into 21 Zernike coefficients (Z0-Z20) following the OSA/ANSI convention. By utilizing these Zernike coefficients, we were able to represent and characterize the wavefront phase. Each coefficient corresponds to a specific property of the wavefront, such as oblique and vertical astigmatism (Z3, Z5), defocus (Z4), trefoil (Z6, Z9), and coma (Z7, Z8). Decomposing the wavefront into Zernike polynomials serves as a featurization step, converting diverse forms of data into numerical representations suitable for basic machine learning algorithms.

- The readability of the paper might increase if the introduction would include a part about the resolution and, as well, something about the physics dependence of the x-ray properties from the FEL settings and how or if these physics dependence has been considered or not.

Regarding the resolution, the information provided above already addresses that aspect.

We have included the key physics explaining how the FEL settings impact the x-ray properties. The central physics that we adjusted is the detuning effect to effectively excite high-order modes. In our case, we over-tapered the undulator to induce the excitation of the donut mode. To provide a detailed explanation of this underlying physics, we have added text in the manuscript around Fig 2.

Indeed, the experiments revealed interesting XFEL physics when certain parameters of the electron bunch and the undulators are varied. In the no-taper case, due to the fact that the electrons are continuously losing energy, the radiation spectrum is skewed toward the red-shift side. For the taper case, the taper was over-tapered to introduce a detuning to set the resonant frequency in the blue-shift side compared to the radiation frequency in the exponential growth region, i.e., before the tapered region. Thus, the microbunching will now support high-order modes according to the dispersion relation discussed below in the Method Section. In our case, the donut mode is excited, as shown in the intensity plot in Fig. 2, while the spectrum shows spectral tails at high energy, as seen in the normalized spectrum plot in Fig. 2

- Figure 1 is very complex and has a long, rather descriptive caption. The authors should consider to simplify the figure and to make the caption more concise. As a suggestion one might consider to remove (d) and (e) and rather make these separate figures. (d) is anyway not really comprehensive.

Due to the journal's guidance on the number of figures, we are unable to include additional figures based on current word count. Fig. 1 serves as an overall introductory figure for the Introduction section, providing background information. The details are provided in the main body of the text and the Methods section. We have cut out the TMO station diagram from figure 1(a) to make the figure more compact.

- Figure 2 shows experimental averaged results of spectra, intensity and phase. Intensity and phase are later not considered, spectrum is shown again in figure 4. I suggest this figure to be moved to the method when describing the experimental results. The contrast could

be better, in particular for the phase plots.

We have replaced the colormap to enhance the contrast, resulting in an improved visual representation.

- On page 5 a very short paragraph describes the relation of Zernike coefficients to wavefronts. This is very short and not comprehensible to the expert. Better extend and add to methods? As well a transfer matrix is described (partially shown in Fig. 1), but nothing is said about the relevance or even uniqueness of this transformation. - Figure 3 is impressive in that the lines almost not differ. However, it is very difficult to read for this very reason. The relevance of the finding, despite being an impressive match, remains unclear as it is not discussed how these coefficients describe the real wavefront.

The wavefront can be effectively represented by projecting it onto a basis of Zernike polynomials, which form a complete orthonormal basis in 2D space. This projection allows us to express the wavefront in terms of Zernike coefficients, which effectively capture the dominant features and aberrations present in the wavefront. Decomposing the wavefront into Zernike polynomials serves as a featurization step, converting diverse forms of data into numerical representations suitable for basic machine learning algorithms.

It is important to note that the matrix shown in Fig. 1 is not the transformation matrix used to calculate the Zernike coefficients. Instead, it serves as an illustration of the correlation between electron parameters and the wavefront output. We have included additional sentences in the Method section to provide a detailed explanation of the relationship between Zernike coefficients and wavefronts.

Highly accurate wavefront measurements were conducted using a Talbot wavefront sensor [34], which has recently been successfully demonstrated with XFEL radiation [21, 22, 23]. The number of Zernike terms required for an accurate representation of a wavefront phase depends on the complexity of the wavefront and the desired level of accuracy. Typically, a small number of Zernike terms (around 10-20) can be used to achieve a reasonable approximation. These Zernike coefficients effectively capture the dominant features and aberrations present in the wavefront, while minimizing the influence of noise. In our specific case, considering both the absolute values and standard deviations of the higher-order Zernike coefficients to be very small compared to the dominant terms, we retrieved the wavefront phase and decomposed it into 21 Zernike coefficients (Z0-Z20) following the OSA/ANSI convention. By utilizing these Zernike coefficients, we were able to represent and characterize the wavefront phase. Each coefficient corresponds to a specific property of the wavefront, such as oblique and vertical astigmatism (Z3, Z5), defocus (Z4), trefoil (Z6, Z9), and coma (Z7, Z8). Decomposing the wavefront into Zernike polynomials serves as a featurization step, converting diverse forms of data into numerical representations suitable for basic machine learning algorithms.

- In figure the marking of part a. and b. needs to be added. In the caption it says: 'small values', but since nowhere absolute values are indicated it is fully unclear what small is.

Either here a relative comparison needs to be made or the absolute value be compared to absolute values of the coefficients.

- On page 8 top it says ‘ The decrease in the variation of the . . . ’, however it is not described in comparison to what, or function of what the values decrease. This would be important for the understanding of this part.

We have added notations (a) and (b) in Fig. 5 to provide clarity. In Fig. 5(a), the ‘small values’ refer to the relatively smaller values of the RMS prediction errors (prediction vs measurement) compared to the standard deviation (SD) of the measurement. This means that the green points are smaller than the blue points in Fig. 5(a). Essentially, it indicates that the shot-to-shot RMS prediction error is lower than the shot-to-shot variation for each Zernike term. Furthermore, both the measurement and prediction exhibit similar SD values (green points and orange points), suggesting that the model captures most of the variations.

Fig. 5(a) displays the SDs and RMS prediction errors for each Zernike term, and we have improved the clarity of the legends. The overall single-shot wavefront RMS error between measurement and prediction is 0.141 rad. In comparison to the shot-to-shot SD of 0.269 rad, the relative prediction accuracy for shot-to-shot variations is approximately 52%.

We have included the following sentences in the Shot-to-shot Variations subsection to describe the shot-to-shot prediction error, variation, and relative accuracy:

Based on the single-shot wavefront phase data, the RMS prediction error between the predicted and measured wavefront phase is determined to be 0.141 rad. Additionally, the standard deviation of the wavefront phase from shot to shot is calculated to be 0.269 rad. Consequently, the estimated relative error for predicting shot-to-shot fluctuations is approximately 52%. Refer to the Methods section on prediction error and accuracy evaluation for further details.

Furthermore, we have added a new subsection titled “Prediction Error and Accuracy Evaluation” in the Method section, which explains how we assess the prediction error, variation, and relative accuracy:

When we have two 2D wavefront phase maps, the RMS difference between these wavefronts can be computed as $\sqrt{\|\Delta\mathbf{X}\|_2^2}$. Here, $\Delta\mathbf{X}$ represents the phase difference within the circular aperture. Alternatively, this difference can be expressed in terms of Zernike coefficients as $\|\Delta\mathbf{Z}\|_2 = \sqrt{\sum_j (\Delta Z_j)^2}$, where ΔZ_j signifies the discrepancy on each Zernike coefficient. This formula is used to calculate the RMS error between the measured wavefront and the predicted wavefront. Additionally, we can assess the shot-to-shot or case-to-case variations by considering ΔZ_j as the standard deviation of each Zernike coefficient. By dividing the RMS prediction error by the standard deviation of the wavefronts, the wavefront prediction error can be evaluated as a relative error.

- Below the rms difference of the single-shot wavefront phase is quoted to be 0.141 rad, while two pages before the seemingly same value is given as 0.138 rad. Why is there a difference

or are these different values? This should be better explained.

These two numbers represent the shot-to-shot variations from the full beam (0.141 rad) and the slotted beam (0.138 rad), which are nearly identical. In order to eliminate confusion, we have opted to report only one number for shot-to-shot variations and one number for case-to-case variations. To provide clarity and avoid confusion, we have included the following sentences:

In the “Analysis of predictions from the slotted electron beam configurations” subsection: The root-mean-square (RMS) prediction error for the average wavefront phases was determined to be 0.0169 rad. Furthermore, the standard deviation of wavefront phase from case to case was found to be 0.236 rad. Based on these values, the estimated relative error for predicting average case-to-case fluctuations is approximately 7%.

In the “Shot-to-shot Variations” subsection: Based on the single-shot wavefront phase data, the RMS prediction error between the predicted and measured wavefront phase is determined to be 0.141 rad. Additionally, the standard deviation of the wavefront phase from shot to shot is calculated to be 0.269 rad. Consequently, the estimated relative error for predicting shot-to-shot fluctuations is approximately 52%.

REVIEWERS' COMMENTS

Reviewer #1 (Remarks to the Author):

In their response to the reviewers, the authors addressed reviewers' comments and made significant additions to the manuscript. This improves the readability of the figures, increases the quality of the presentation of methodology and results and makes the the work in general more sound.

Considering the implemented corrections and reviewers' comments, I recommend the manuscript for publication in its current version.

Reviewer #2 (Remarks to the Author):

I think the authors appropriately answered the comments and questions from the reviewers and the manuscript was revised well. The authors clearly showed the ability to predict the XFEL characteristics from the electron beam properties for each shot. The descriptions of the ANN structure and the theoretical background were also added. I think the readers can understand the authors' work more clearly and I understand the novelty of the study. Of course, there are already many papers that report applications of ANN to particle accelerators or light sources. However, I think it is the first result to predict the XFEL characteristics from the electron beam information shot-to-shot.

Reviewer #3 (Remarks to the Author):

The authors have responded very well to the reviewers remarks and the article has gained in readability and accessibility.

I recommend publication in the present form.

Response to the reviewers

We appreciate the time and effort the reviewers have invested in evaluating our work again and appreciate their recommendations for publication.

Reviewer #1 (Remarks to the Author):

In their response to the reviewers, the authors addressed reviewers' comments and made significant additions to the manuscript. This improves the readability of the figures, increases the quality of the presentation of methodology and results and makes the work in general more sound.

Considering the implemented corrections and reviewers' comments, I recommend the manuscript for publication in its current version.

Reviewer #2 (Remarks to the Author):

I think the authors appropriately answered the comments and questions from the reviewers and the manuscript was revised well. The authors clearly showed the ability to predict the XFEL characteristics from the electron beam properties for each shot. The descriptions of the ANN structure and the theoretical background were also added. I think the readers can understand the authors' work more clearly and I understand the novelty of the study. Of course, there are already many papers that report applications of ANN to particle accelerators or light sources. However, I think it is the first result to predict the XFEL characteristics from the electron beam information shot-to-shot.

Reviewer #3 (Remarks to the Author):

The authors have responded very well to the reviewers remarks and the article has gained in readability and accessibility.

I recommend publication in the present form.